# Preliminary Study of ADHD Biomarkers in Adults with Focus on Serum Iron and Transcranial Sonography of the Substantia Nigra

**DOI:** 10.3390/ijerph18094875

**Published:** 2021-05-03

**Authors:** Geon-Ho Bahn, Sang-Min Lee, Minha Hong, Seung-Yup Lee

**Affiliations:** 1Department of Psychiatry, Kyung Hee University College of Medicine, Seoul 20447, Korea; mompeian@khu.ac.kr (G.-H.B.); rheegyatso@gmail.com (S.-Y.L.); 2Department of Psychiatry, Myongji Hospital, Hanyang University College of Medicine, Goyang 10475, Korea; npmhhong@gmail.com

**Keywords:** transcranial sonography, stimulant, iron, ferritin, attention-deficit/hyperactivity disorder, substantia nigra, biomarkers, adults

## Abstract

As previous studies have reported abnormalities in the iron indices of peripheral blood and hyperechogenicity of the substantia nigra (SN) in children and adolescents with attention-deficit/hyperactivity disorder (ADHD), we aimed to examine the same in adults with ADHD using transcranial Doppler sonography (TCS). In addition, we compared the iron indices and TCS findings before and after methylphenidate (MPH) treatment. A total of 39 participants aged ≥19 years (13 patients and 26 healthy controls) were recruited from Kyung Hee University Hospital between October 2018 and September 2019. All subjects were clinically evaluated based on the ADHD diagnostic criteria in the DSM-5, the Adult ADHD Self-Report Scale, and the Diagnostic Interview for ADHD in Adults (DIVA-5). Further, the iron indices including serum iron, ferritin, and mean platelet volume were determined. Additionally, TCS focused on the midbrain and echogenicity of the SN was conducted. Follow-up for all items was conducted for five ADHD patients after MPH treatment. Patients with ADHD had significantly lower education levels (number of years) than controls. There were no statistically significant differences in serum iron indices or the echogenic area between ADHD and control groups. Further, there were no significant changes in iron indices or TCS findings after MPH medication. Unlike previous studies, this study showed no differences between patients with ADHD and controls. Therefore, it is important to determine if these null findings were due to different target populations (children vs. adults) or other factors, including ADHD subtypes.

## 1. Introduction 

There is considerable variation in the persistence rate of attention-deficit/hyperactivity disorder (ADHD) from childhood to adulthood, with estimates ranging from 15% [1] to 27.8% [2], depending on the definition of ADHD. It is more difficult to diagnose ADHD in adults than in children and adolescents due to the high prevalence of other physical and mental disorders [3]. Therefore, newer and more reliable biological diagnostic methods are urgently needed to confirm the diagnosis of ADHD, especially in adults. 

Iron has been implicated in brain function and dopaminergic activity, and abnormalities in iron levels have been used to explain the etiology of ADHD [4]. Given the involvement of iron in neurocognition and behaviors, it is unsurprising that iron deficiency has been reported in children with problems such as a lack of concentration and hyperactivity [5]. Iron deficiency is associated with a decreased expression of dopamine transporter (DAT), and abnormalities in DAT expression are known to be related to genetic vulnerabilities in ADHD [6]. Wang et al. [4] recommended the use of multiple indices of iron status in addition to ferritin and serum iron levels for more accurate results when exploring the relationship between iron levels and ADHD incidence. Mean platelet volume (MPV) has recently been used as an indicator of systemic inflammatory response in children with ADHD [7]. Iron levels in the brain may be more indicative of ADHD pathophysiology than peripheral or systemic iron levels [8]. A meta-analysis showed that thalamic iron concentration in children with ADHD was significantly lower than that in typical children [6].

Although magnetic resonance imaging (MRI) can be used for the measurement of brain iron content [8], transcranial Doppler sonography (TCS) is also a simple, inexpensive, and non-invasive neuroimaging method [9]. TCS measures the absorption coefficient of the deep brain structure in the intact skull, which can be used to evaluate cellular iron homeostasis in distinct brain areas, especially the substantia nigra (SN), as well as central dopamine conditions [9]. To the best of our knowledge, two previous studies have used TCS to examine the SN in pediatric patients with ADHD—one study evaluated hyperechogenicity of the SN in children aged 7–16 years with ADHD [10], and another study evaluated the correlation of SN echogenicity with symptoms of inattention, hyperactivity, and impulsivity in children and adolescents with ADHD [11]. 

As previous studies using TCS have mostly focused on children and adolescents with ADHD, this study targeted adults with ADHD to determine the correlation between the iron indices of peripheral blood and echogenicity of the SN using TCS. Further, we examined the changes in biological indices after psychostimulant treatment.

## 2. Materials and Methods 

### 2.1. Participants 

This prospective case–control study was conducted between October 2018 and September 2019. Thirteen patients aged ≥19 years who were newly diagnosed with ADHD were enrolled during their initial visit at Kyung Hee University Hospital. Subjects in the ADHD group were drug naïve or had not used anti-ADHD drugs in the previous 3 months. Age- and sex-matched healthy controls were recruited through offline and online portals at Kyung Hee University Hospital. 

All participants were evaluated by clinicians (GHB and SYL) using the Adult ADHD Self-Report Scale, Korean version (K-ASRS) and the Diagnostic Interview for ADHD in Adults (DIVA-5). ADHD diagnosis was based on the Diagnostic and Statistical Manual of Mental Disorders, 5th edition [12]. The exclusion criteria for subjects were as follows: chronic disease affecting iron metabolism, history of serious neurological diseases or head trauma that could affect brain imaging studies, and mental disorders including intellectual disabilities. 

### 2.2. Measures

#### 2.2.1. K-ASRS 

The ASRS is an 18-item self-report scale based on the DSM-IV symptom criteria developed by the World Health Organization [13], which was translated and validated to develop the Korean version, K-ASRS [14]. The scale is composed of two parts, part A and part B, with a total of 18 questions answered using a five-point Likert scale ranging from 0 (never) to 4 (very often). Part A, the ASRS screener, comprises six questions, and a subject with positive responses for at least four out of the six items is considered to be at an elevated risk for ADHD [15]. The sensitivity and specificity of part A of the K-ASRS were 0.627 and 0.804, respectively [16]. In this study, subjects replied to part A of the K-ASRS only.

#### 2.2.2. DIVA-5 

The DIVA is a semi-structured interview tool for diagnosing ADHD in adults, which is available from the DIVA foundation website [17]. DIVA-5 has been previously translated into Korean and validated [18]. This tool consists of three parts that cover the following themes: (1) ADHD symptoms in childhood and adulthood; (2) age of ADHD onset; and (3) areas of impairment due to ADHD. If three or more criteria were met for either inattention and/or hyperactivity/impulsivity in childhood before the age of 12 years, and five or more criteria were met in adulthood as reported by the patient and collateral informant(s), the requirements for a clinical diagnosis of lifetime ADHD were met [18].

#### 2.2.3. Iron Indices

The following indices related to iron metabolism were used: serum iron, ferritin, transferrin, total iron-binding capacity (TIBC), hemoglobin, mean corpuscular volume (MCV), mean corpuscular hemoglobin (MCH), mean corpuscular hemoglobin concentration (MCHC), and mean platelet volume (MPV). 

#### 2.2.4. TCS

We used a color-coded duplex ultrasound system equipped with a 2.0–2.5 MHz phased array transducer (HDI 5000, SonoCT: Philips). Using a low frequency of 2.0–2.5 MHz makes it easier for ultrasound to pass through the temporal bone in the skull. Ultrasound parameters included a penetration depth of 16 cm and a dynamic range of 45–50 dB. The examination with the TCS probe was performed through the preauricular acoustic bone window, on both the right and left sides. The visualization of the target structures may depend on the quality of the temporal acoustic bone window. The SN was identified as a tie-shaped echogenic signal within the butterfly-shaped mesencephalic brainstem. To check hyperechogenicity, the sonographer marked areas manually but measurements were performed automatically (Figure 1). TCS results were examined by a well-trained sonographer who was not informed about the participants’ group allocation. All TCS images were stored and evaluated offline by an experienced neurologist who had no information about the participants’ group allocation. 

#### 2.2.5. Clinical Global Impression-Severity (CGI-S) and CGI-Improvement (CGI-I) 

The CGI-S is a seven-step scale for measuring the severity of symptoms (1 = no disorder to 7 = severe disorder) and the CGI-I is a seven-step scale for measuring improvements in symptoms (1 = impressive improvement, 4 = no improvement, and 7 = strong deterioration) [19]. 

### 2.3. Methods 

Once the subjects were finalized, blood tests were performed to measure iron composition indicators followed by TCS. For the ADHD group, methylphenidate (MPH) treatment was initiated if a patient opted for it, and no placebo was administered to the control group. In the ADHD group, for patients receiving MPH, blood tests and TCS were performed again when the CGI-I indicated “very much” or “much improved” (CGI-I ≤ 2) [19]. The principal investigator evaluated the CGI-S and CGI-I scores. 

Written informed consent was obtained from all subjects prior to their participation in the study. The study protocol was approved by the Institutional Research Board of Kyung Hee University Hospital (KMC IRB 2018-02-007), and followed the ethical standards of the Declaration of Helsinki.

### 2.4. Statistical Analysis

The comparison of the demographic characteristics, iron indices, and TCS findings between the ADHD and control groups was performed using the Wilcoxon rank-sum test and the chi-square test. A multiple generalized linear model was used to obtain results which were adjusted for age and education level. The comparison before and after MPH treatment in patients who underwent follow-up tests was performed using the Wilcoxon signed-rank test. SAS 9.4 (SAS Institute Inc., Cary, NC, USA) was used for statistical analysis, and the significance level was set at *p* < 0.05. 

## 3. Results

There were 13 subjects (5 women, 38%) in the ADHD group and 26 subjects (13 women, 50%) in the control group. 

Patients with ADHD had significantly lower education levels (*p* = 0.001) than controls (Table 1). Additionally, when education level was stratified by sex, both men (*p* = 0.0095) and women (*p* = 0.0024) in the ADHD group had lower education levels than those in the control group.

There were no statistically significant differences in sex, age, iron indices (serum iron, ferritin, transferrin, TIBC, hemoglobin, MCV, MCH, MCHC, and MPV), or TCS findings (midbrain area (right and left) and echogenic area of the SN (right and left)) between the two groups (Table 1). In addition, there were no significant findings for any items between the groups when adjusted for sex. 

Among the 13 subjects in the ADHD group, five subjects had to be excluded from the follow-up tests because of early dropout and three subjects were excluded because they were prescribed additional psychotropic medication during the study period due to depression, mood swing, or insomnia. Finally, five subjects took the MPH medication only and completed the follow-up tests (Table 2). No significant changes were observed in iron indices or TCS findings when results from before and after MPH medication were compared (Table 3).

## 4. Discussion

In this study, none of the iron-related indicators showed a significant difference between the ADHD and control groups, regardless of sex. These results were not consistent with a previous meta-analysis of iron status in children with ADHD [4]. However, another previous study using two large population-based birth cohorts did not report an association between peripheral ferritin concentrations and ADHD symptoms [20]. On the other hand, the results from the limited number of studies in adults with ADHD do not corroborate the results discussed here. In a study on the association between restless legs syndrome (RLS) and iron deficiency in adults with ADHD, iron deficiency was more common in patients with ADHD and RLS than in controls [21].

As variations in systemic iron levels in ADHD patients have been reported, recent studies have suggested that iron levels in the brain, rather than systemic iron levels, may be associated with the pathophysiology of ADHD in children [6]. A meta-analysis showed that lower thalamic iron concentrations were found in children with ADHD compared to healthy controls [6]. Although examining brain iron content using MRI may be a more reliable biomarker of ADHD, its application has several drawbacks including its high cost and the difficulty in using it for pediatric patients [6].

The influence of comorbid psychiatric and physical conditions may be one of the reasons why the results of this study differed from those of previous studies. Although the clinician interviewed all participants, it may have been difficult to exclude both physical and neurological diseases. Serum iron levels can be influenced by several conditions, including inflammation, obesity, Tourette syndrome [22], and epilepsy [23]. Adult women with only ADHD symptoms and those with both ADHD and obsessive compulsive disorder (OCD) symptoms showed significantly reduced ferritin concentrations compared to men with only ADHD symptoms or those with both ADHD and OCD symptoms. This may suggest overlapping dopaminergic and serotoninergic dysfunctions in ADHD, OCD, and RLS [24]. 

Youth with ADHD showed a significantly larger SN echogenic size than healthy control children [10,11]. Krauel et al. [11] reported a more marked hyperechogenic SN size in the ADHD children than in the control group. When the results were stratified by sex, the difference in the SN echogenicity between boys only in the ADHD group and those in the control group was more pronounced than the difference between the total samples of the two groups. However, Romanos et al. [10] found that sex did not affect this parameter. In this study, the size of the midbrain and echogenic area in the SN did not significantly differ between the ADHD and control groups, regardless of sex. We could not find any study related to the changes in size of the midbrain before and after stimulant medication in adult patients with ADHD. In Parkinsonism research, there are sufficient studies on the echogenicity of the SN in patients. In ADHD, an increase in age has been associated with an increase in the size of the hyperechogenicity in the SN, whereas an increase in age of Parkinsonism patients has been associated with a reduction in the size of the hyperechogenicity in the SN [25]. However, Parkinsonism is different from ADHD, and subjects with Parkinsonism are mostly elderly. 

There is not enough data on the association between ADHD medication and brain function or structural changes, especially in adult patients. While evidence suggests that iron supplementation reduces the severity of ADHD symptoms, brain imaging data before and after iron supplementation are insufficient [26]. It has been reported that there is a marked reduction in the initially elevated striatal DAT expression with a low dose of methylphenidate, as measured by single-photon emission computed tomography [27]. Although youth with ADHD may have less prominent age-related elevation of iron levels in the brain than that seen in typical development, long-term use of psychostimulant medications may compensate through the normalizing effect on iron levels in the basal ganglia [8].

In this study, there were no significant differences in the echogenic area in the SN in follow-up tests conducted after MPH treatment. Similar studies cannot be found in the literature, which limits the interpretation of our results. Moreover, polypharmacy including ADHD drugs, is frequently prescribed for adult ADHD patients due to comorbid psychiatric and medical conditions [3]. Therefore, it is difficult to evaluate only the effects of ADHD drugs and only the effects in the midbrain. A meta-analysis on activation likelihood estimation indicated that ADHD pathophysiology may involve network interactions rather than just regional abnormalities [28]. In this study, the lack of differences in the neuroimaging findings of TCS in the follow-up may be explained by network interactions rather than region-specific changes. Further, it is necessary to confirm whether these changes were due to the effect of MPH or some other unknown variable. 

This preliminary study has several limitations. First, the results of this study, especially the follow-up tests in patients receiving medications, should be very carefully interpreted considering the different dosages, durations, and types of medications, and the small number of subjects with no control group. Second, although TCS is convenient to use, there are some disadvantages in using this technique—the signal and noise of TCS are influenced by individual differences and the results may vary depending on the proficiency of the sonographer. Currently, the results from TCS or iron-related laboratory tests are not sufficient and are inconsistent for use as biomarkers for ADHD, and cannot be applied to clinical practice. Nonetheless, future research should yield valuable results based on this study design through the recruitment of sufficient numbers of patients and appropriate control of the medications used, as well as the use of control groups. 

Although the results of this study contradict the results of previous studies and did not demonstrate any significant difference in the echogenicity of the SN between patients with ADHD and controls, this is the first study to examine the echogenicity of the SN in adults with ADHD and the effects of psychostimulant treatment on the midbrain and SN. 

## 5. Conclusions

The serum iron indices and the neuroimaging findings of TCS reported in this study did not corroborate with those reported in previous studies involving children and adolescents. However, this study included adult subjects. Therefore, it is necessary to confirm the findings of this study using larger samples. Further, the role of intracerebral networks related to the iron pathway in the pathophysiology of ADHD in adults should also be explored. 

## Figures and Tables

**Figure 1 ijerph-18-04875-f001:**
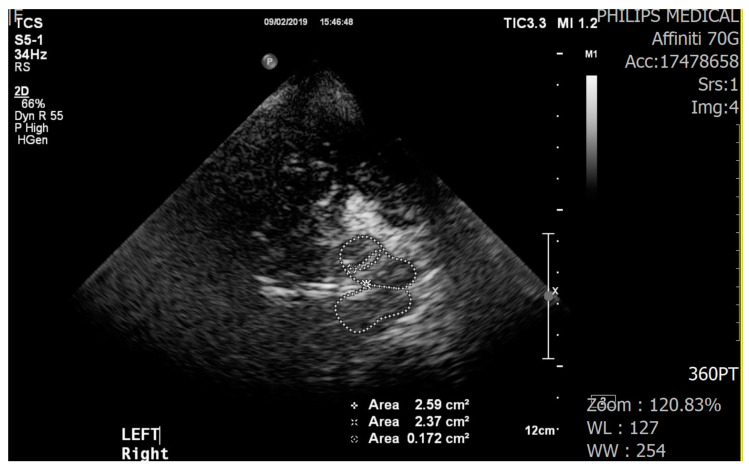
Transcranial sonography images of the midbrain (butterfly-shaped, left and right) and hyperechogenicity of the substantia nigra (encircled by the dotted line, inside the midbrain).

**Table 1 ijerph-18-04875-t001:** Demographic and clinical characteristics of attention-deficit/hyperactivity disorder (ADHD) and healthy control groups.

Items		Control (*n* = 26)	ADHD (*n* = 13)	*p* *	p ^†^
Sex	Male	13	8	0.4956	
	Female	13	5		
Age (years)		27.29 ± 2.86	29.32 ± 8.01	0.8823	
Education (years)		17.50 ± 1.24	15.31 ± 0.95	0.0001	
Serum iron (µg/dL)		113.69 ± 30.5	134.85 ± 52.82	0.1976	0.1129
Ferritin (ng/mL)		85.94 ± 59.87	160.14 ± 224.66	0.6259	0.8843
Transferrin (mg/dL)		249.73 ± 38.17	268.08 ± 35.04	0.1158	0.3187
TIBC (µg/dL)		322.54 ± 46.25	339.77 ± 46.75	0.1783	0.4306
Hemoglobin (g/dL)		14.23 ± 1.35	14.34 ± 1.69	0.6466	0.2192
MCV (fL)		90.90 ± 2.98	90.61 ± 3.98	0.8014	0.9636
MCH (pg)		30.34 ± 1.03	30.42 ± 1.68	0.6679	0.8884
MCHC (g/dL)		33.38 ± 0.83	33.57 ± 0.79	0.4874	0.7958
MPV (fL)		7.57 ± 0.75	7.75 ± 0.54	0.2819	0.5871
Midbrain area, right (cm^2^)		2.74 ± 0.40	2.61 ± 0.36	0.4345	0.1544
Midbrain area, left (cm^2^)		2.73 ± 0.40	2.62 ± 0.36	0.4698	0.2353
Echogenic area, SN, right (cm^2^)		0.07 ± 0.13	0.03 ± 0.08	0.4246	0.1608
Echogenic area, SN, left (cm^2^)		0.08 ± 0.11	0.04 ± 0.07	0.2281	0.1624

* Wilcoxon rank-sum test, p ^†^: adjusted by age and education with multiple generalized linear model, serum iron: (reference level 60~180), ferritin: (13~150), transferrin: (200~360), hemoglobin: (12~16), TIBC: total iron-binding capacity (215~535), MCV: mean corpuscular volume (81~99), MCH: mean corpuscular hemoglobin (27~31), MCHC: mean corpuscular hemoglobin concentration (33~37), MPV: mean platelet volume (7.2~11.1), SN: substantia nigra.

**Table 2 ijerph-18-04875-t002:** List of subjects examined before and after the methylphenidate treatment.

Number	Sex	Age (Years)	Medication	Follow-Up Interval (Weeks)	CGI-S	CGI-I
1	Male	47.3	MPH OROS 18 mg qd	21	5	2
2	Male	22.1	MPH OROS 36 mg qd	20	5	1
3	Male	31.6	MPH IR 5 mg bid	23	6	2
4	Male	22.3	MPH ER 20 mg qd	18	5	2
5	Female	40.4	MPH OROS 36 mg qd	13	6	2

qd: once a day, bid: twice a day, MPH: methylphenidate, OROS: osmotic release oral system, IR: immediate release, ER: extended release, CGI-S: clinical global impression-severity, CGI-I: clinical global impression-improvement.

**Table 3 ijerph-18-04875-t003:** Comparison of findings from laboratory and transcranial Doppler sonography (TCS) before and after methylphenidate treatment (*n* = 5).

Items	Baseline	With Medication	Diff		*p* *
Serum iron (µg/dL)	126.60 ± 60.81	90.60 ± 53.51	−36.00 ± 66.34	−25.00(64.00)	0.3125
Ferritin (ng/mL)	176.54 ± 125.86	120.00 ± 103.04	−56.54 ± 124.33	−2.60(215.10)	0.6250
Transferrin (mg/dL)	257.80 ± 43.07	258.80 ± 77.73	1.00 ± 38.99	−5.00(26.00)	0.6250
TIBC (µg/dL)	317.60 ± 61.66	342.40 ± 93.62	24.80 ± 41.88	15.00(38.00)	0.3125
Hemoglobin (g/dL)	13.88 ± 2.42	14.46 ± 1.82	0.58 ± 1.01	0.30(0.40)	0.2500
MCV (fL)	90.80 ± 4.20	89.72 ± 4.07	−1.08 ± 1.68	−1.50(2.40)	0.3125
MCH (pg)	30.70 ± 2.22	30.40 ± 1.94	−0.30 ± 0.66	−0.30(1.10)	0.5000
MCHC (g/dL)	33.78 ± 1.14	33.66 ± 1.11	−0.12 ± 0.29	0.00(0.30)	0.6250
MPV (fL)	8.12 ± 0.55	7.84 ± 0.34	−0.28 ± 0.75	−0.30(0.20)	0.4375
Midbrain area, right (cm^2^)	2.84 ± 0.27	2.66 ± 0.30	−0.18 ± 0.09	−0.18(0.09)	0.0625
Midbrain area, left (cm^2^)	2.83 ± 0.29	2.67 ± 0.30	−0.16 ± 0.10	−0.18(0.08)	0.0625
Echogenic area, SN, right (cm^2^)	0.00 ± 0.00	0.00 ± 0.00	0.00 ± 0.00	0.00(0.00)	
Echogenic area, SN, left (cm^2^)	0.00 ± 0.00	0.06 ± 0.12	0.06 ± 0.12	0.00(0.00)	1.0000

* Wilcoxon rank-sum test, TIBC: total iron-binding capacity, MCV: mean corpuscular volume, MCH: mean corpuscular hemoglobin, MCHC: mean corpuscular hemoglobin concentration, MPV: mean platelet volume, SN: substantia nigra.

## Data Availability

The data that support the findings of this study are available from the corresponding author, upon reasonable request.

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
