# Peer review of "Preliminary Study of ADHD Biomarkers in Adults with Focus on Serum Iron and Transcranial Sonography of the Substantia Nigra"

_ijerph, 2021, doi:10.3390/ijerph18094875_

Round 1
Reviewer 1 Report
Paper titled (Preliminary study of biomarkers of ADHD in adults: focused on serum iron and transcranial sonography on substantia nigra) by Bahn et al discussed avery interesting topic and tested the possible association between some bio/other markers and ADHD in adults.
I recommend if the authors highlight these items before publication:
1- Statistical analysis: Kindly justify why you used Wilcoxon rank some test or chi square test?? was this right?
Kindly mention the type of your data in each marker (quantitative with normal dist, quantitative with non normal dist, ordinal data....etc?) and how you reached this decision. Based on the type of data we choose the appropriate stat test. This is very important and critical.
From a preliminary evaluation, most of data seems not in normal distribution
2- Based on comment (1): kindly revise data for ferritin n Table 1, I notice a large difference between the 2 groups.
3-In key words: please add (adults)
4- Please mention in the study limitations or future directions that more studies are warranted to perform a similar study on children with ADHD
Author Response
Reviewer 1
Paper titled (Preliminary study of biomarkers of ADHD in adults: focused on serum iron and transcranial sonography on substantia nigra) by Bahn et al discussed a very interesting topic and tested the possible association between some bio/other markers and ADHD in adults.
I recommend if the authors highlight these items before publication:
Statistical analysis: Kindly justify why you used Wilcoxon rank some test or chi square test?? was this right?
- The Wilcoxon rank sum test was applied to adjust age and sex in Table 1 data analysis, and the chi square test was used to compare blood tests related to iron components.
Kindly mention the type of your data in each marker (quantitative with normal dist, quantitative with non-normal dist, ordinal data....etc?) and how you reached this decision. Based on the type of data we choose the appropriate stat test. This is very important and critical. From a preliminary evaluation, most of data seems not in normal distribution.
- As a basis for determining whether the data related to iron components are normal or abnormal, I described the reference range of the Kyung Hee University Hospital laboratory as legends in Table 1 for the following items related with iron:
* wilcoxon rank sum test, p†: adjusted by age and education with multiple generalized linear model, Serum iron: (reference level 60~180), Ferritin: (13~150), Transferrin: (200~360), Hemoglobin: (12~16), TIBC: total iron binding capacity (215-535), MCV: mean corpuscular volume (81-99), MCH: mean corpuscular hemoglobin (27-31), MCHC: mean corpuscular hemoglobin concentration (33-37); MPV: mean platelet volume (7.2-11.1), SN: substantia nigra.
Based on comment (1): kindly revise data for ferritin n Table 1, I notice a large difference between the 2 groups.
- As you mentioned, although the difference in Ferritin concentration between the two groups shows big different numbers, it was not statistically significant after re-checking with the statistician.
In key words: please add (adults)
- As you recommended, “adults” was added to the key words, which is an important subject of this study.
Please mention in the study limitations or future directions that more studies are warranted to perform a similar study on children with ADHD
- "Future research expects valuable results based on this study design through sufficient number of patient recruitment, children and adult patients, and appropriate control of the medications used, as well as control groups."
In addition, I corrected several errors in this draft. For example, there was an error in the first sentence: There is a considerable variation in the persistence rate prevalence of attention-deficit/hyperactivity disorder (ADHD) from childhood to adulthood, with estimates ranging from 15% [1] to 27.8% [2] to 65% [2].

Reviewer 2 Report
Overall, the manuscript is an original contribution, and data address a relevant issue not well examined previously Introduction The authors need to add more citations about the iron deficiency relationships with children with problems (lack of concentration and hyperactivity). This idea is a central idea for your study. I suggest adding citations for this affirmation "Since previous studies have mostly focused on children and adolescents with ADHD." Material and method The authors must provide more information about the sample in the participants' section, like age and gender information. A description of the socio-economic status of the sample is also required. It's unnecessary to describe part B of the K-ASRS due to participants not answering this section. For the self-reported scales, I suggest adding examples of the ítems. In the methods section, this part is not clear "treatment was initiated if a patient opted for it" how many did not opt for it? Please explain better this part of the method. The authors must provide more details about the ethical concerns of the study. Discussion The implications of the study results must be described in more detail.Author Response
Reviewer 2
Overall, the manuscript is an original contribution, and data address a relevant issue not well examined previously
Introduction
The authors need to add more citations about the iron deficiency relationships with children with problems (lack of concentration and hyperactivity). This idea is a central idea for your study. I suggest adding citations for this affirmation "Since previous studies have mostly focused on children and adolescents with ADHD."
- As mentioned in the introduction, there are only two articles studying ultrasound images of substantia nigra using TCS in ADHD patients, and the motive of this study is that the subjects in the previous studies were children and adolescents. Therefore, it was not easy to add more references. Even though the reference is harder to find, but your opinion is very valuable, I revised the first sentence of the leading paragraph as follows: "Since previous studies using TCS have mostly focused on children and adolescents with ADHD."
In addition, I also corrected several errors in this draft. For example, there was an error in the first sentence in ‘Introduction’: There is a considerable variation in the persistence rate prevalence of attention-deficit/hyperactivity disorder (ADHD) from childhood to adulthood, with estimates ranging from 15% [1] to 27.8% [2] to 65% [2].
Material and method
The authors must provide more information about the sample in the participants' section, like age and gender information.
- Table 1 presents the average age of patients and control groups and the ratio of men and women. Table 2 lists the gender and age of five patients who performed the second test after methylphenidate treatment.
A description of the socio-economic status of the sample is also required.
- While authors aimed at identifying biologic markers related to iron pathway in this study, we did not include the socioeconomic status. As you pointed out, while reviewing the findings again, we recognized the possibility that the economic status would be an important variable that affects the patients’ condition, in addition to the patients’ ADHD symptoms. Subsequent studies should include socioeconomic factors such as economic power and marital status.
It's unnecessary to describe part B of the K-ASRS due to participants not answering this section.
- As you commented, part B description was deleted. Instead, the K-ASRS description was modified as follows:
The scale is composed of two parts, Part A and B, with a total of 18 questions answered using a five-point Likert scale ranging from 0 (never) to 4 (very often).
For the self-reported scales, I suggest adding examples of the ítems.
- Since ASRS is a tool developed by the World Health Organization and published on its websites, most mental health officials can download from that website, therefore, there may be no need to demonstrate the items of ASRS
In the methods section, this part is not clear "treatment was initiated if a patient opted for it" how many did not opt for it? Please explain better this part of the method.
- As indicated in part in the “Results” section, five out of 13 adults were diagnosed as ADHD and dropped out early without medication. Another three took other psychotropics besides methylphenidate for sleep and/or mood problems. Finally, only five patients who underwent secondary examinations took only methylphenidate as a treatment drug.
The authors must provide more details about the ethical concerns of the study.
- I added the contents related to ethical concern to the existing IRB paragraph as follows:
“Written informed consent was obtained from all subjects prior to their participation in the study. The study protocol was approved by the Institutional Research Board of Kyung Hee University Hospital (KMC IRB 2018-02-007), and followed the ethical standards of the Helsinki declaration.
Discussion
The implications of the study results must be described in more detail.
- Currently, the results from TCS or iron-related laboratory tests were not sufficient and inconsistent to use as biomarkers for ADHD, and can’t be applied to clinical practice. Nonetheless, research in the future expects valuable results based on this study design through sufficient number of patient recruitment and appropriate control of the medications used, as well as control groups.

Reviewer 3 Report
The work can be very interesting and well designed, but it has the disadvantage that it seems more like a good project, but it is very scarce in patients, so it does not seem that conclusions can be drawn from it. Vomo preliminary study is fine, but I think the authors should continue to collect data in order to give greater scientific rigor to their possible conclusions. I am uncertain whether the publication in the current format could prevent a publication of the style with a greater number of patients in the near future or could, on the contrary, serve as a stimulus for it. In this sense, I believe that a future plan should be discussed in which the authors indicate their future projection in the recruitment of patients and in the follow-up of drug-treated patients.
Author Response
Reviewer 3
The work can be very interesting and well designed, but it has the disadvantage that it seems more like a good project, but it is very scarce in patients, so it does not seem that conclusions can be drawn from it. Vomo preliminary study is fine, but I think the authors should continue to collect data in order to give greater scientific rigor to their possible conclusions.
I am uncertain whether the publication in the current format could prevent a publication of the style with a greater number of patients in the near future or could, on the contrary, serve as a stimulus for it. In this sense, I believe that a future plan should be discussed in which the authors indicate their future projection in the recruitment of patients and in the follow-up of drug-treated patients.
- You pointed out exactly what I was worried about while writing this paper. As you have commented, I hesitated if it was right to publish it as it is because there were few subjects and I was unsure of the results of the data analysis. However, as you wrote, there are very few studies using TCS for patients with ADHD, especially adults, therefore, I decided to submit this draft because I thought I could find meaning as a trigger to subsequent research. As you advised, I added the following future plans at the end of this draft. "Future research expects valuable results based on this study design through sufficient number of patient recruitment and appropriate control of the medications used, as well as control groups."

Round 2
Reviewer 1 Report
Thanks for the authors fro addressing the comments
Reviewer 3 Report
The authors have adequately qualified the comments I made in the previous review, for this reason and as preliminary results I believe that the present work can be published